# Imaging Characterization of Non-Rheumatoid Retro-Odontoid Pseudotumors: Comparison with Atlantoaxial Manifestation of Rheumatoid Arthritis

**DOI:** 10.3390/medicina58091307

**Published:** 2022-09-19

**Authors:** You-Seon Song, In-Sook Lee, Kyoung-Hyup Nam, Dong-Hwan Kim, In-Ho Han, Hwangbo Lee, Yeon-Joo Jeong, Jeong-A Yeom

**Affiliations:** Department of Radiology, Pusan National University Hospital, 179, Gudeok-ro, Busan 49241, Korea

**Keywords:** retro-odontoid pseudotumor, rheumatoid arthritis, MR imaging, atlantoaxial joint

## Abstract

*Background**and Objectives*: To date, imaging characterization of non-rheumatic retro-odontoid pseudotumors (NRROPs) has been lacking; therefore, NRROPs have been confused with atlantoaxial joint involvement of rheumatoid arthritis (RA). It is important to differentiate these two disease because the treatment strategies may differ. The purpose of this study is to characterize imaging findings of NRROPs and compare them with those of RA. *Material and Methods*: From January 2015 to December 2019, 27 patients (14 women and 13 men) with NRROPs and 19 patients (15 women and 4 men) with RA were enrolled in this study. We evaluated various imaging findings, including atlantoaxial instability (AAI), and measured the maximum diameter of preodontoid and retro-odontoid spaces with magnetic resonance imaging (MRI) and computed tomography (CT). *Results*: Statistical significance was considered for *p* < 0.05. AAI was detected in eight patients with NRROPs and in all patients with RA (*p* < 0.0001). Seventeen patients with NRROPs and six patients with RA showed spinal cord compression (*p* = 0.047). Compressive myelopathy was observed in 14 patients with NRROPs and in 4 patients with RA (*p* = 0.048). Subaxial degeneration was observed in 25 patients with NRROPs and in 9 patients with RA (*p* = 0.001). Moreover, C2-3 disc abnormalities were observed in 11 patients with NRROPs and in 2 patients with RA (*p* = 0.02). Axial and longitudinal diameter of retro-odontoid soft tissue and preodontoid and retro-odontoid spaces showed significant differences between NRROP and RA patients (*p* < 0.0001). Furthermore, CT AAI measurements were differed significantly between NRROP and RA patients (*p* < 0.05). *Conclusions*: NRROPs showed prominent retro-odontoid soft tissue thickening, causing compressive myelopathy and a high frequency of subaxial and C2-3 degeneration without AAI.

## 1. Introduction

Retro-odontoid pseudotumor (ROP) is a non-neoplastic, soft tissue thickening adjacent to the dens of cervical 2 (C2) vertebra [1]. It is an uncommon but important imaging finding, as it causes neurologic symptoms and can be a marker of systemic disease.

Although the etiology and pathophysiology of ROP formation and growth are unclear, it can be secondary to various causes [2]. The most common causes are rheumatoid arthritis (RA), acute trauma, and craniocervical junction malformation [1,2]. Historically, ROPs have been classified as rheumatoid or non-rheumatoid in etiology [1]. When associated with RA, pseudotumors may be referred to as pannus [1]. Histopathologically, ROPs present with various compositions depending on the etiology, suggesting that there may be variability in the underlying processes, leading to the proliferation of soft tissue in each condition. [1]. In RA, pannus formation is attributed to inflammation of the synovial membrane, resulting in overgrowth of hyaline cartilage and periarticular tissue formation [3]. To date, the recognition of non-rheumatic ROPs (NRROPs) in the field of radiology has been lacking; therefore, NNROPs have been confused with atlantoaxial joint involvement of the RA pannus. Thickening of retro-odontoid soft tissue can be observed using magnetic resonance imaging (MRI) or computed tomography (CT) in adults with both RA and non-RA etiologies [4]. However, it is expected that the degree and pattern of atlantoaxial joint involvement in NRROPs and RA will differ. It is important to differentiate between these two etiologies because the treatment plans will differ.

To the best of our knowledge, there are no distinguishing imaging features that can reliably differentiate rheumatoid from non-rheumatoid etiologies in ROPs. Thus, the aim of this study was to evaluate the imaging findings of NRROPs compared with those of RA pannus in the atlantoaxial joint.

## 2. Materials and Methods

This study was approved by the institutional review board, and informed consent was waived due to the retrospective nature of the study.

### 2.1. Patients

From January 2015 to December 2019, we searched for cases with mass effect or soft tissue thickening in the retro-odontoid region using radiologic reports on the picture archiving communication system and medical chart reviews.

In this study, NRROP was defined as a mass-like soft tissue thickening in the retro-odontoid space without related systemic illness, generalized joint disease, or calcified components on CT [5]. Patients with clinical or laboratory findings of RA or a history of head or neck trauma were excluded. All patients had negative serum rheumatoid factor test results. We also investigated cases of atlantoaxial manifestation of RA during the same period. All patients with RA were confirmed according to the 2010 American College of Rheumatology/European League Against Rheumatism classification criteria.

Ultimately, 27 patients (14 women and 13 men; mean age, 71.1 ± 13.4 years; age range, 43–99 years) with NRROPs and 19 patients (15 women and 4 men; mean age, 60.5 ± 16.1 years; age range, 17–79 years) with RA were enrolled in this study. Through medical chart review, the patients’ symptoms, treatment methods, surgery, and postoperative symptom relief were investigated.

### 2.2. Imaging Study

All patients underwent MRI, CT, and plain radiographs of the cervical spine, including the atlantoaxial joint.

The MRI protocol of the cervical spine was as follows. MRI of the spine was performed using a 1.5-or 3-T system with a spinal surface or phased-array coil. T1-weighted and T2-weighted turbo spin-echo images were obtained in the sagittal and axial planes. The slice thickness was 3 mm in the axial plane and 2 mm in the sagittal plane, with a 0.3 mm interval.

The protocol for cervical spine CT was as follows. CT scans were performed using multidetector CT units (SMS Definition or SMS Definition AS+, Siemens Healthineers, Erlangen, Germany) with 64 or 128 detector rows. The imaging parameters were 120 kVp, 60–500 mAs, 3–5 mm slice thickness, and a 0.4–0.5 mm interval. Sagittal, coronal, and axial reformatted multiplanar images were obtained using transverse images with a section thickness of 2 mm. Contrast-enhanced images were not obtained. The scan times ranged from 10 to 18 s. The scan range started from the skull base to the upper thoracic spine, including at least the T1 level.

Plain radiographs included anteroposterior, lateral, flexion, and extension views.

### 2.3. Image Evaluation

Atlantoaxial instability (AAI) or subluxation was defined as an atlantodental interval (ADI) > 4 mm on flexion and extension lateral radiographs according to the criteria described by White and Panjabi [6].

On spine MRIs, we evaluated the presence or absence of spinal cord compression, compressive myelopathy (high signal intensity within the spinal cord on T2-weighted image), erosion of the odontoid process, ossification of the posterior or anterior longitudinal ligament, disc or bony degeneration below the C3 (subaxial disc or bony degeneration), and herniation or degeneration of the C2-3 disc. Signal intensities of retro-odontoid soft tissue or pannus were not evaluated, as they were non-specific in both patient groups.

The long- and short-axis diameters of retro-odontoid soft tissue thickening were measured using sagittal and axial T2-weighted MR images. The length of the long axis was measured in the sagittal T2WI, where soft tissue thickening was the longest (Figure 1A). The length of the short axis was measured from the posterior cortex of the odontoid process to the apex of the thickened soft tissue at the position where soft tissue thickening was the most prominent in the axial T2WI (Figure 1B).

To assess the extent of soft tissue lesions in the atlantoaxial joint, the preodontoid space (maximum distance between the anterior arch of C1 and dense C2) and the retro-odontoid space (maximum distance between the dense of C2 and the apex of the retro-odontoid soft tissue) were measured on sagittal T2-weighted MR images (Figure 2A).

For evaluation of lateral and anteroposterior atlantoaxial joint instability, we measured the right and left distances between the odontoid process and the lateral mass of C1, as well as the central distance between the anterior arch of C1 and the odontoid process of C2 at the C1-2 level, where the anterior arch of C1 and the odontoid process of C2 were shown side by side on axial CT scans. When asymmetrically scanned, the larger of each measured value in the continuous image was included in the result. (Figure 3B,C).

### 2.4. Statistical Analysis

Dichotomous and categorical data are reported as numbers (percentages). Continuous data are reported as means ± standard deviation and range. A chi-square test was used to evaluate differences between patients with NRROPs and RA. Age and other measurements, including retro-odontoid soft tissue thickness, pre- and retro-odontoid spaces on MR images, and atlantoaxial instability on CT images, were also compared between patients with ROPs and RA using a two-sample t-test. Analysis was performed using SPSS software (version 21; IBM, Armonk, NY, USA). Statistical significance was accepted for *p* values < 0.05.

## 3. Results

All 27 patients with NRROPs complained of symptoms, including tingling sensation, numbness, or bilateral or unilateral arm weakness of varying degrees. A total of 12 of the 19 RA patients reported variable degrees of neck pain, and 7 reported radiating arm pain or numbness.

The clinical data and imaging results of NRROP and RA patients with atlantoaxial joint involvement are summarized in Table 1. Patient age, atlantoaxial instability, spinal cord compression, compressive myelopathy, C2-3disc degeneration or herniation, and disc or bony degeneration below C3 showed significant differences between patients with NRROPs and RA (*p* < 0.05). Patients with NRROPs were older and exhibited a higher rate of spinal cord compression (*n* = 17, 63%), compressive myelopathy (*n* = 14, 52%), degeneration or herniation of the C2-3 disc (*n* = 11, 41%), and subaxial disc or bony degeneration (*n* = 25, 93%) compared with those with RA (Figure 1 and Figure 2). In contrast, atlantoaxial instability (*n* = 19, 100%) was significantly higher in patients with RA (Figure 3). Erosion of the odontoid process did not differ significantly between the two groups.

The measurements of retro-odontoid soft tissue thickness, pre- and retro-odontoid space on MR images, and atlantoaxial instability on CT images are summarized in Table 2. On MR images, the axial and longitudinal diameters of the retro-odontoid soft tissue and pre- and retro-odontoid spaces showed significant differences between patients with NRROPs and RA (*p* < 0.05). The axial and longitudinal diameters of the retro-odontoid soft tissue and retro-odontoid spaces were larger in patients with NRROPs than those with RA. The pre-odontoid space was wider in patients with RA than those with NRROPs.

On CT images, the lateral distance of the left and right atlantoaxial joints between the odontoid process and lateral mass of C1 showed a significant difference between patients with RA and NRROPs (*p* < 0.05); however, the difference between the right and left distance was not significant between the two groups (*p* = 0.706).

## 4. Discussion

According to our study, NRROPs resulted in more prominent retro-odontoid soft tissue thickening than RA involvement of the atlantoaxial joint. NRROPs showed a high incidence of compressive myelopathy and subaxial and C2-3 degeneration and a low incidence of AAI.

There is currently no standard that defines the point at which retro-odontoid soft tissue thickness is considered an ROP, although reports of abnormal retro-odontoid soft tissue typically describe a thickness of more than 3 mm [7,8]. Tojo et al. [8] measured retro-odontoid soft tissue thickness in 503 consecutive patients. They showed that the mean retro-odontoid soft tissue thickness in patients with RA was 3.7 mm; however, only 14% of the study patients had RA. This result is similar to those obtained in our study, which showed that the mean retro-odontoid soft tissue thickness in RA patients (5.3 mm) was lower than that of in NRROP patients. In general, ROPs may cause cord compression in radiologic studies [9]. However, ROP formation does not always cause cord compression. In our study, cord compression was observed in 17 patients (63%) with NRROPs and in 6 patients (32%) with RA. Additionally, cord signal changes suggesting myelopathy were observed in 14 (52%) cases with NRROPs and in 4 (21%) cases with RA.

According to several previous reports, in cases of NRROPs, most patients are elderly men [10,11]. However, in our study, most NRROP patients were women. In addition, the mean age of patients with non-rheumatoid ROPs was higher than that of patients with RA.

Although most ROPs in patients with RA have been associated with AAI [7,12,13], NRROPs have been reported in patients with and without AAI in several case reports or series [12,14,15]. In this study, AAI was associated with approximately 30% of patients with NRROPs; however, all patients with RA showed AAI. In addition, AAI was prominent in the anterior-to-posterior direction in RA and on the lateral sides in NRROP patients.

Ryu et al. [9] identified a negative correlation between retro-odontoid soft tissue thickness and ADI in patients with RA. In our study, the pre-odontoid space was significantly increased in patients with RA, and the retro-odontoid space was significantly increased in patients with NRROPs.

The formation and growth of NRROPs have been correlated with degenerative alterations of the cervical spine or underlying instability of the atlantoaxial joint, which is responsible for continuous mechanical stress on the ligamentous and cartilaginous complex of the atlantoaxial joint [4,8,12,15,16]. According to a review of previously reported cases [11,12,16], most patients with NRROPs presented with moderate or severe osteoarthritic changes at the atlantoaxial joint. In addition, Barbagallo et al. [17] hypothesized that extensive subaxial cervical spine spondylotic changes might be coupled with compensatory hypermobility in the C1-2 area and that such a phenomenon would eventually induce the formation of ROPs. Thus, moderate or severe osteoarthritis of C1-2 or loss of mobility in the middle and lower cervical segments, resulting in excessive motion of the C1-2 segment, transfers mechanical stress to the atlantoaxial joint, resulting in repeated tear and repair of the ligaments, including the posterior longitudinal ligament and the transverse ligament, inducing fibrocartilaginous metaplasia and fibrovascular ingrowth and resulting in the development of ROPs [11,12,16,18]. In our study, the incidence of C2-3-disc degeneration and subaxial disc or bony degeneration was significantly higher in patients with NRROPs than in those with RA. Thus, most of our patients with NRROPs had limited mobility in the subaxial cervical segments. Moreover, nine patients with NRROPs had a history of subaxial surgery. In contrast, previous investigations described ROPs as retrodental disc masses or migration of a C2-3-sequestered disc [19]. In the current study, only 11 NRROP cases (41%) showed C2-3-disc degeneration or herniation.

Diffuse idiopathic skeletal hyperostosis (DISH) may also contribute to compensatory hypermobility at the craniocervical junction, potentially leading to ROP formation [20]. Jun et al. [21] reported a case of ROP with DISH. In contrast, Chikuda et al. [22] found that extensive ossification of the anterior longitudinal ligament (OALL) was highly prevalent in patients with ROPs. They suggested that extensive OALL and ankyloses of the adjacent segments are risk factors for the formation of pseudotumors. The altered biomechanics of the upper cervical spine due to ankylosis might load mechanical stress on the atlantoaxial joint, leading to the development of an ROP without AAI [5]. However, in our study, we observed focal thickening or ossification of the PLL in some patients with NRROPs (15%) and RA (11%). However, no patient in our study presented with definite DISH. Moreover, OALL was observed in neither group.

In RA, an inflamed and thickened synovium (pannus) generally occurs around the odontoid process of dens and causes bone erosion [23]. However, the frequency of odontoid process erosion was did not significantly differ between patients with NRROPs and those with RA in this study.

Treatment strategies may differ depending on the etiology of the ROP. According to a 2015 meta-analysis [24], over the last 50 years, there has been a 27% decrease in the prevalence of AAI in patients with RA if the disease was controlled with medications, such as disease-modifying antirheumatic drugs (DMARDs). Therefore, it is important to distinguish between NRROPs and RA. Both NRROPs and RA can be treated by decompression through surgery. Surgical treatments include lesion resection, C1-2 fixation, or a combination of these procedures. Multiple reports have demonstrated regression of ROPs after surgical stabilization of the craniocervical junction, both in patients with and without RA, obviating the need for direct transoral removal of the compressing mass [8,13,14,17]. In our study, 8 of the 27 patients with NRROPs and 7 patients with RA underwent posterior fixation of the atlantoaxial joint and showed improved symptoms and a progressive decrease in ROPs.

Pathologic findings reported in a previous study included fibrocartilage, fibroproliferative changes, and partial fibrin deposits [18]. There is some controversy regarding the nature of this tissue. Some authors [13,25] suggest that this tissue is an inflammatory granulation tissue, whereas others have suggested that it is a reactive fibrous tissue secondary to mechanical stress. In our study, pathological examination of three patients with NRROPs with partial resection of the retro-odontoid soft tissue showed degeneration of fibrocartilage tissue. There appears to be a limit to the ability to determine the exact mechanism of pseudotumor generation using pathologic results alone.

This study is subject to some limitations. First, the number of patients included in the study was small due to the uncommon nature of atlantoaxial joint involvement in patients with RA and NRROPs. Second, only 3 of 15 patients who underwent surgery in our study underwent pathologic examination for retro-odontoid soft tissue thickening or hypertrophy, as surgical access to the retro-odontoid area was challenging and hazardous in most cases. However, there were no problems associated with diagnosis of these two groups based on imaging and clinical findings.

In conclusion, atlantoaxial manifestations of RA and NRROPs are clinically and radiologically distinct. NRROPs are characterized as a local change caused by a degenerative process, whereas the atlantoaxial manifestation of RA is caused by a systemic disease. NRROPs exhibit prominent retro-odontoid soft tissue thickening, causing compressive myelopathy and a high frequency of subaxial and C2-3 degeneration without AAI. Awareness of these characteristic imaging findings of NRROPs may facilitate their differentiation from RA and aid in treatment planning.

## Figures and Tables

**Figure 1 medicina-58-01307-f001:**
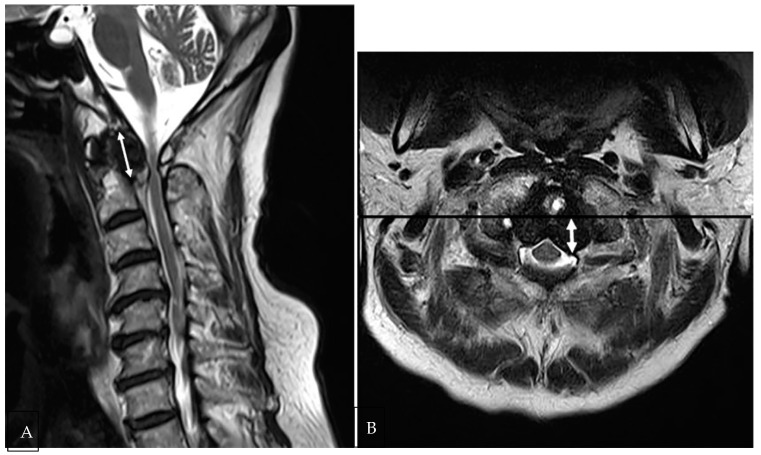
A 78-year-old female patient with a non-rheumatoid retro-odontoid pseudotumor. (**A**) Mass-like hypertrophy of retro-odontoid soft tissue on a sagittal T2-weighted MR image. The long axis of soft-tissue thickening (double-pointed arrow) was obtained by measuring the longest length. Erosion of the odontoid process is noted. (**B**) The short-axis of retro-odontoid soft-tissue hypertrophy (double-pointed arrow) was measured from the posterior cortex of the odontoid process to the apex of the thickened soft tissue at the position where the soft-tissue thickening was the most prominent in the axial T2WI.

**Figure 2 medicina-58-01307-f002:**
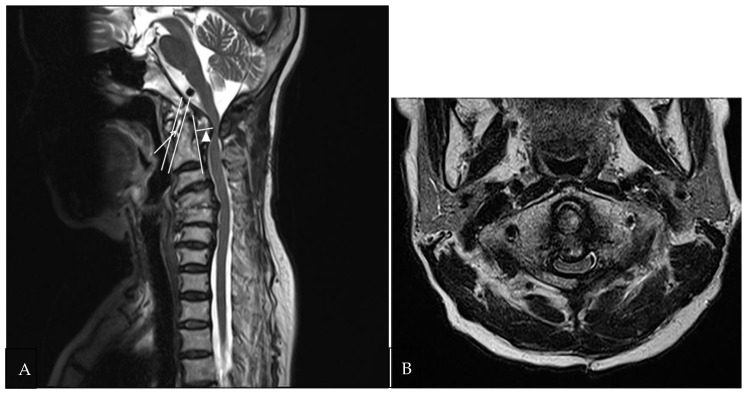
A 75-year-old female patient with a non-rheumatoid retro-odontoid pseudotumor. (**A**) Preodontoid space (arrow) was measured from the anterior arch of C1 and dense C2 on a sagittal T2WI. The retro-odontoid space (arrowhead) was measured from the posterior cortex of the odontoid process to the protrusion of the hypertrophied soft tissue lesion. (**B**) The spinal cord is compressed due to hypertrophy of retro-odontoid soft tissue. The high signal intensity inside the cord suggests compressive myelopathy on axial T2WI.

**Figure 3 medicina-58-01307-f003:**
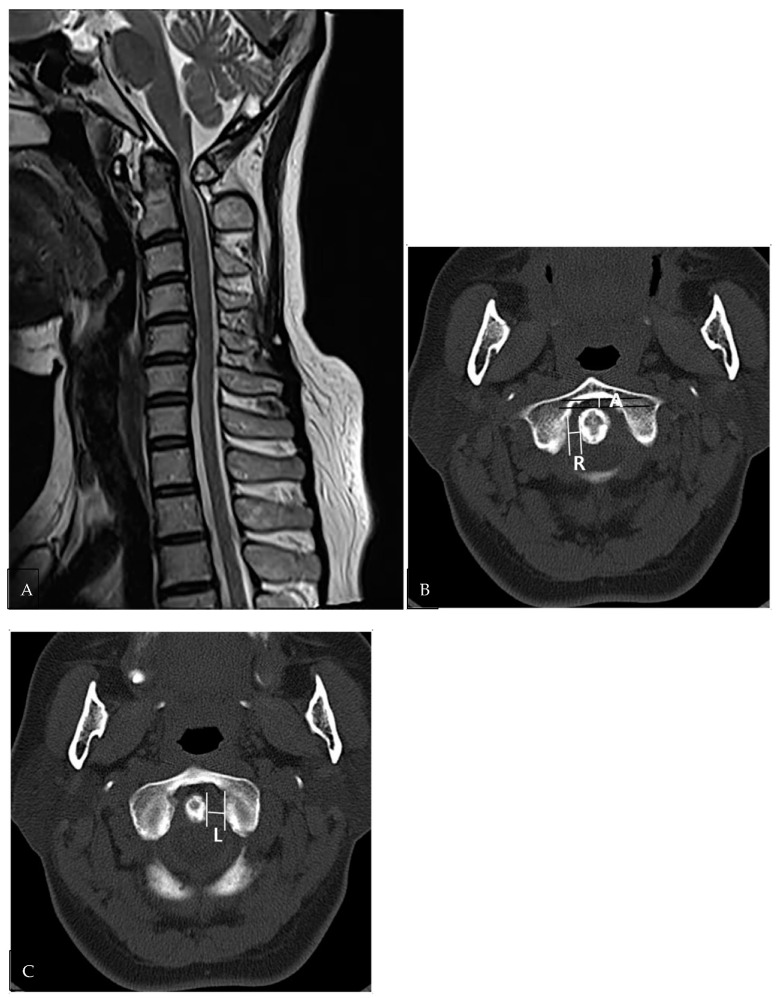
A 58-year-old female patient with rheumatoid arthritis. (**A**) The atlantoaxial joint space represents abnormal widening on sagittal T2WI. The spinal cord of the C1-2 level demonstrates compression and myelopathy, although retro-odontoid soft tissue shows mild thickening. (**B**,**C**) Each right (R in (**B**)) and left gap (L in (**C**)) between the odontoid process and lateral mass of C1, as well as the central gap (A in (**B**)) between the anterior arch of C1 and the odontoid process of C2 was evaluated for lateral and anteroposterior atlantoaxial joint subluxation. The measurement was performed in the slice at the C1-2 level at the anterior arch of C1, and the odontoid processes of C2 are shown side by side and were measured using axial computed tomography (CT) scans.

**Table 1 medicina-58-01307-t001:** Clinical data and imaging results of ROP patients and RA patients with atlantoaxial joint involvement.

	Non-Rheumatoid Retro-Odontoid Pseudotumor (Total = 27)	Rheumatoid Arthritis (Total = 19)	*p*-Value
Sex	F:M = 14:13	F:M = 15:4	0.082
Age	71.1 ± 13.4 (43–99)	60.5 ± 16.1 (17–79)	0.022
Atlantoaxial subluxation	Y:N = 8:19 (30%)	Y:N = 19:0 (100%)	<0.0001
Spinal cord compression	Y:N = 17:10 (63%)	Y:N = 6:13 (32%)	0.047
Compressive myelopathy	Y:N = 14:13 (52%)	Y:N = 4:15 (21%)	0.048
Erosion of odontoid process	Y:N = 12:15 (44%)	Y:N = 14:5 (74%)	0.065
Ossification or thickening of PLL	Y:N = 4:23 (15%)	Y:N = 2:17 (11%)	0.636
Ossification or thickening of ALL	Y:N = 0:27 (0%)	Y:N = 0:19 (0%)	NA
Subaxial disc or bony degeneration	Y:N = 25:2 (93%)	Y:N = 9:10 (47%)	0.001
C2-3 disc degeneration or herniation	Y:N = 11:16 (41%)	Y:N = 2:17 (12%)	0.02

Note. *p*-value < 0.05 considered statistically significant. PLL = posterior longitudinal ligament, ALL = anterior longitudinal ligament, F = female, M = male, Y = yes, N = no, NA = not applicable.

**Table 2 medicina-58-01307-t002:** Measurements of retro-odontoid soft-tissue thickness and pre- and retro-odontoid space on MRIs and atlantoaxial instability on CT images.

	Retro-Odontoid Pseudotumor (*n* = 27)	Rheumatoid Arthritis (*n* = 19)	*p*-Value
Thickening of retro-odontoid soft tissue on MRIs			
Axial diameter (mm)	10.05 ± 3.47 (3.2–17.4)	4. 68 ± 1.64 (2.0–8.3)	<0.0001
Longitudinal diameter (mm)	23.53 ± 6.04 (14.1–46.7)	16.99 ± 3.42 (12.19–24.1)	<0.0001
Pre-odontoid space (mm)	2.78 ± 1.49 (1.16–7.6)	5.54 ± 2.17 (3.1–11.1)	<0.0001
Retro-odontoid space (mm)	9.25 ± 3.25 (4.1–16.0)	5.19 ± 1.76 (2.8–10)	<0.0001
Atlantoaxial instability on CT images			
Right gap	2.63 ± 1.5 (0.9–6.5)	4.66 ± 1.88 (2.6–8.4)	0.001
Central gap	2.1 ± 1.15 (0.67–4.7)	5.18 ± 2.03 (2.4–10.1)	<0.0001
Left gap	2.87 ± 1.63 (0.9–7.93)	4.51 ± 1.88 (1.6–7.2)	0.005
Difference between bilateral sides	1.07 ± 1.22 (0.1–4.7)	1.2 ± 1.09 (0.2–4.8)	0.706

Note. *p*-value < 0.05 considered statistically significant.

## Data Availability

The datasets used and/or analyzed during the current study are available from the corresponding author upon reasonable request.

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
