# Peer review of "Imaging Characterization of Non-Rheumatoid Retro-Odontoid Pseudotumors: Comparison with Atlantoaxial Manifestation of Rheumatoid Arthritis"

_medicina, 2022, doi:10.3390/medicina58091307_

Round 1

Reviewer 1 Report

This study performed medical image analysis and measurements to compare non-rheumatoid retro-odontoid pseudotumor (NRROP) and rheumatoid arthritis (RA) pannus. There are several concerns in my general comments:

1. The writing is kind of arbitrary as a scientific paper, and largely impede the reading of your paper. I suggest that the authors should carefully proofread the paper thoroughly before resubmission.

2. The sample sizes in this study are so small to achieve statistical significance.

3. Most importantly, the authors did not clearly elaborate the clinical significance of this study. That is, why should NRROP be distinguished from RA pannus?

Below are my detailed comments for the consideration of the authors.

ABSTRACT

L11, “Background”: please briefly introduce why should NRROP be distinguished from RA?

L11-12: Please define ROP and RA when they occurred first.

L17, “Results”: What do these p-values mean?

L17-25: These statistical differences described in the Results do not make sense to me, as the numbers of NRROP (27) and RA (19) are too biased. Also, their sample sizes in this study are so small.

L24: Why does not AAI be introduced at Line 18?

Please note that typically, abbreviations should be avoided in the Abstract. In the main body, all abbreviations that occur first should be defined.

INTRODUCTION

L31: What is the definition of pseudotumor in this paper? Just mean soft tissue thickening?

L34: AAI should be introduced at Line 32. Please carefully check to other abbreviations to improve the legibility of your paper.

L40-41: According to the context (as described in the Abstract, Line 11), ROP is specifically referred to as non-rheumatoid. But the description here seems conflicting; why is ROP caused by RA?  

L48-55: Throughout the introduction, the authors did not clearly state the clinical significance of this study. It is unclear why NRROP should be distinguished from RA pannus. Are there dreadful consequences if NRROP is confused with RA pannus? Please expand the introduction to clarify this.

L56-59: What is the hypothesis of this study that you want to test?

MATERIALS AND METHODS

L87: The thickness (5 mm) and interval (5 mm) of CT slices are so large. Was the resolution sufficient to diagnose or measure? For example, I don't think that it allows to examine if the ADI > 4 mm.

L92-95: Similar to CT, please describe MRI slice thicknesses and intervals, as you used MRI to measure soft tissue thickenings.

L136-137: What the difference of the “central distance” from the pre-odontoid space (at Line 120)? If they are the same, please consider using the same terminology.

L158: What are other measurements? Please list.

RESULTS

L163-177: It is confusing why surgical outcomes were reported. Are these surgical outcomes related to the objective of this study? If so, the procedures that the patients underwent should be firstly introduced in the Methods.

L190 and L204: Please introduce the logistic regression in the Methods.

DISCUSSION

The first paragraph of the Discussion should summarize important results, and check if they support or reject your hypothesis.

L211-212: The definition of ROP should be introduced in the Introduction.

L218-220: It is confusing. Please double-check that ROP causes cord compression, as cord compression was observed 63% patients with NRROP.

L225: Please remind me what measurements are related to AAI. Did you measure range of motion at C1-2 (e.g., please refer to https://pubmed.ncbi.nlm.nih.gov/34390949/)? Or, AAI is just represented by anatomic dimensions, e.g., the ADI?

L234-236: I feel that this should be introduced in the Introduction.

L235-237: Soft tissue thickening can maintain stability?

L258-264: I did not find relevant data regarding C2-3 in the Results. If you did measure them, please add them in the Results.

L303-304: Is this a conclusion from the measurements in this study?

Author Response

Thank you for your interest in my article. I am very pleased to get the reviewers’ comments about my manuscript, “Imaging characterization of non-rheumatoid retro-odontoid pseudotumor; comparison with atlantoaxial manifestation of rheumatoid arthritis”. I’m honored to have a chance to revise my manuscript for the journal, Medicina. I paid attention to all criticisms by the reviewers and amended my manuscript accordingly. I also highlighted the changes I made in the manuscript by using tracking changes.

Reviewer 1

English language and style

(x) Extensive editing of English language and style required
( ) Moderate English changes required
( ) English language and style are fine/minor spell check required
( ) I don't feel qualified to judge about the English language and style

Yes

Can be improved

Must be improved

Not applicable

Does the introduction provide sufficient background and include all relevant references?

( )

( )

(x)

( )

Are all the cited references relevant to the research?

( )

(x)

( )

( )

Is the research design appropriate?

( )

(x)

( )

( )

Are the methods adequately described?

( )

( )

(x)

( )

Are the results clearly presented?

( )

( )

(x)

( )

Are the conclusions supported by the results?

( )

(x)

( )

( )

Comments and Suggestions for Authors

This study performed medical image analysis and measurements to compare non-rheumatoid retro-odontoid pseudotumor (NRROP) and rheumatoid arthritis (RA) pannus. There are several concerns in my general comments:

  1. The writing is kind of arbitrary as a scientific paper, and largely impede the reading of your paper. I suggest that the authors should carefully proofread the paper thoroughly before resubmission.

 à Thank you for your comment.

    I totally agree with your opinion. So we proofread the paper and made some changes which is more suitable for scientific paper.

  1. The sample sizes in this study are so small to achieve statistical significance.

 à Thank you for your comment. As you mentioned, the sample sizes in this study are small. However, the disease in this study is somewhat rare (including NRROP and atlantoaxial involvement of RA) so the sample size may be small. So we added about this problem in the limitation section as follows.

“This study has some limitations. First, the number of patients included in this study was small. This is due to atlantoaxial joint involvement being uncommon in patients with RA and NRROP.”

  1. Most importantly, the authors did not clearly elaborate the clinical significance of this study. That is, why should NRROP be distinguished from RA pannus?

 à Thank you for your comment. We agree with your opinion. We changed the introduction part to make clear about that.

Below are my detailed comments for the consideration of the authors.

ABSTRACT

L11, “Background”: please briefly introduce why should NRROP be distinguished from RA?

à We added the reason why NRROP should be distinguished from RA on background section as follows.

“It is important to differentiate these two disease because the treatment strategies may differ.”

L11-12: Please define ROP and RA when they occurred first.

à Thank you for your comment. We defined them when they occurred first on the abstract. 

L17, “Results”: What do these p-values mean?                                                               

à These p-values are the result from the statistical anaylsis and it is significant when they are less than 0.05. And those factors are significant to differentiate NRROP and RA in this study.

L17-25: These statistical differences described in the Results do not make sense to me, as the numbers of NRROP (27) and RA (19) are too biased. Also, their sample sizes in this study are so small.

 à Thank you for your comment. As you mentioned, the sample sizes in this study are small. However, the disease in this study is somewhat rare (including NRROP and atlantoaxial involvement of RA) so the sample size may be small. So we added about this problem in the limitation section as follows.

“This study has some limitations. First, the number of patients included in this study was small. This is due to atlantoaxial joint involvement being uncommon in patients with RA and NRROP.”

L24: Why does not AAI be introduced at Line 18?

 àThank you for your comment. We added that in Line 18.

Please note that typically, abbreviations should be avoided in the Abstract. In the main body, all abbreviations that occur first should be defined.

INTRODUCTION

 àThank you for your comment. We added the definition of pseudotumor in this study as follows.

“Retro-odontoid pseudotumor (ROP) is a non-neoplastic soft tissue thickening adjacent to the dens of cervical 2 (C2) vertebra.”

We didn’t use exact distance (ex. More than 3mm) because there is no clear definition about this. This is discussed in discussion section.

L34: AAI should be introduced at Line 32. Please carefully check to other abbreviations to improve the legibility of your paper.

à Thank you for your comment. We carefully proofread the manuscript and checked abbreviations.

L40-41: According to the context (as described in the Abstract, Line 11), ROP is specifically referred to as non-rheumatoid. But the description here seems conflicting; why is ROP caused by RA?  

à Thank you for your comment. We agree with your opinion. There are some confusing sentences in the introduction part. So we revised the introduction section according to your opinion.

 L48-55: Throughout the introduction, the authors did not clearly state the clinical significance of this study. It is unclear why NRROP should be distinguished from RA pannus. Are there dreadful consequences if NRROP is confused with RA pannus? Please expand the introduction to clarify this.

à Thank you for your comment. ROP can be caused by rheumatoid or non-rheumatoid etiology. But it is difficult to distinguish these two by imaging because both of them affect same anatomical location (atlantoaxial joint) and similar imaging findings. But it is important to differentiate them because the treatment stategy is different each other. So we revised the introduction section according to your comment.

L56-59: What is the hypothesis of this study that you want to test?

àThank you for your comment. We tried to test the imaging findings that can be used to differentiate NRROP and RA pannus. Since the treatment strategy will be differ according to the etiology. And also we tried to evaluate the imaging findings of this rare disease entity because there were not so many studies about that.

MATERIALS AND METHODS

L87: The thickness (5 mm) and interval (5 mm) of CT slices are so large. Was the resolution sufficient to diagnose or measure? For example, I don't think that it allows to examine if the ADI > 4 mm.

àThank you for your comment. There was mistake in writing, and sorry about that. The resolution was sufficient to diagnosis and measure. We revised the wrong number.

L92-95: Similar to CT, please describe MRI slice thicknesses and intervals, as you used MRI to measure soft tissue thickenings.

  • We added the slice thickness and intervals of MRI as 3mm thickness, 0.3mm intervals.

L136-137: What the difference of the “central distance” from the pre-odontoid space (at Line 120)? If they are the same, please consider using the same terminology.

  • Thank you for your comment. Actually the measurement method is somewhat different in “central distance” and “pre-odontoid space” in this study. Pre-odontoid space is measured on sagittal MR image and central distance or central gap is measured on axial CT image.

L158: What are other measurements? Please list.

  • Thank you for your comment. We added the list of other measurement.

RESULTS

L163-177: It is confusing why surgical outcomes were reported. Are these surgical outcomes related to the objective of this study? If so, the procedures that the patients underwent should be firstly introduced in the Methods.

  • Thank you for your comment. As you mentioned, surgical outcomes are not important in this study. So we removed about that in the result.

L190 and L204: Please introduce the logistic regression in the Methods.

  • Thank you for your comment. We described statistical analysis in the Methods.

DISCUSSION

The first paragraph of the Discussion should summarize important results, and check if they support or reject your hypothesis.

  • Thank you for your comment. We added the summarization of important result of this study.

L211-212: The definition of ROP should be introduced in the Introduction.

  • The definition of ROP was introduced in introduction section. However, this paragraph is about thickness of retro-odontoid soft tissue and it is related to the result of this study. So we inserted in the discussion section. If it is problematic, please let us know and we will make change.

L218-220: It is confusing. Please double-check that ROP causes cord compression, as cord compression was observed 63% patients with NRROP.

  • Thank you for your comment. According to the degree of soft tissue thickening, ROP may or may not cause cord compression. Not all NRROP or RA pannus cause cord compression, we meant.

L225: Please remind me what measurements are related to AAI. Did you measure range of motion at C1-2 (e.g., please refer to https://pubmed.ncbi.nlm.nih.gov/34390949/)? Or, AAI is just represented by anatomic dimensions, e.g., the ADI?

  • Thank you for your comment. In our study, AAI was evaluated in dynamic plain radiograph (flexion and extension view) according to the criteria described by White and Panjabi

L234-236: I feel that this should be introduced in the Introduction.

  • Thank you for your comment. We removed that part in the discussion.

L235-237: Soft tissue thickening can maintain stability?

  • Thank you for your comment. As you mentioned, it can cause confusion, so we removed that.

L258-264: I did not find relevant data regarding C2-3 in the Results. If you did measure them, please add them in the Results.

  • We evaluated the presence or absence of C2-3 disc degeneration and it is shown in the result section and Table 1.

 L303-304: Is this a conclusion from the measurements in this study?

  • This is the conclusion of this study that more likely imaging findings of NRROP compared to RA pannus.

Reviewer 2 Report

I recommend compacting the discussion part and point out the consequences for treatment

Author Response

I recommend compacting the discussion part and point out the consequences for treatment

  • Thank you for your commnent. We removed some part of the discussion and tried to make it more compact. And we added the consequences for treatment in the discussion section as follows.

“Treatment strategies may differe according to the etiology of the ROP. According to the meta-anaylsis in 2015 [24], over the last 50 years, there has been a 27% decrease in the prevalence of AAI in patients with RA, if the disease control was done with medications such as disease-modifying anti-rheumaticdrugs (DMARDs). Therefore, it is important to distinguish NRROP and RA.”

Round 2

Reviewer 1 Report

Thanks the authors for responding to my comments. Now this paper looks better.